# Automated technique for high-pressure water-based window cleaning and accompanying parametric study

Youngjoo Lee☯, Daesung Kwon☯, Changmin Park☯, Myoungjae Seo☯, TaeWon Seo ⬥ *☯

School of Mechanical Engineering, Hanyang University, Seoul, Korea

☯ These authors contributed equally to this work.
* taewonseo@hanyang.ac.kr

## Abstract

The maintenance of buildings has become an important issue with the construction of many high-rise buildings in recent years. However, the cleaning of the outer walls of buildings is performed in highly hazardous environments over long periods, and many accidents occur each year. Various robots are being studied and developed to reduce these incidents and to relieve workers from hazardous tasks. Herein, we propose a method of spraying high-pressure water using a pump and nozzle, which differs from conventional methods. The cleaning performance parameters, such as water pressure, spray angle, and spray distance, were optimized using the Taguchi method. Cleaning experiments were performed on window specimens that were contaminated artificially. The cleaning performance of the proposed method was evaluated using the image-evaluation method. The optimum condition was determined based on the results of a sensitive analysis performed on the image data. In addition, the reaction force due to high pressure and impact force on the specimens were investigated. These forces were not sufficient to affect the propeller thrust or cause damage to the building's surface. We expect to perform field tests in the near future based on the output of this research.

## 1. Introduction

With the development of architectural technology, many high-rise buildings have been constructed worldwide in recent years. As a result, maintenance tasks such as the cleaning of outer walls present problems with regard to cost and worker safety. The working environment is exceptionally hazardous because workers are suspended either on gondolas or by ropes from very tall buildings.

Many exterior wall-cleaning robots have been studied [1–7] and developed to reduce the cost of cleaning and to relieve workers from hazardous tasks. For example, CSIC's Tito [8], IPC Eagle's HighRise [9, 10], and SkyPro [11, 12] are commercially available with certain product lineups, as shown in Fig 1. These robots are transported up and down buildings by using cables and winches. These commercial robots use typical equipment to clean the exterior walls of buildings, such as water nozzles, squeegees, and brushes [8–12]. Although these devices are

**Data Availability Statement:** All relevant data are within the paper.

**Funding:** (T. Seo) This research was supported by the National Research Foundation of Korea (NRF) Grant funded by the Ministry of Science and ICT for First-Mover Program for Accelerating Disruptive

Technology Development (Management: NRF-2018M3C1B9088328 (Specification No.1: 2018M3C1B9088330, Specification No.2: 2018M3C1B9088331, Specification No.3: 2018M3C1B9088332)). The funders had no role in study design, data collection and analysis, decision to publish, or preparation of the manuscript.

**Competing interests:** The authors have declared that no competing interests exist.

**Fig 1.** Various exterior wall cleaning robots (a) Tito [8] (b) IPC EAGLE [9, 10] (c) SKYPRO [11, 12] (d) Gecko [12–15].

highly effective for cleaning, their use results in uncleaned zones. In the process of traversing optimized moving paths, which prevent collision of the devices with obstacles, uncleaned zones are generated near obstacles such as façades. Another popular robot is Gecko [13–15], which was developed using a highly effective vacuum suction pad [16–18]. However, the applicability of these robots are limited owing to the shapes of the buildings and the decorative fittings on them. The motion speeds are extremely low when vacuum pads are used; therefore, long cleaning times result. In addition, there are limitations on the payload required for the cleaning device.

Several studies on high-pressure cleaning have been conducted to enhance the cleaning performance through spray optimization. Zhang reported that the shape characteristics of the nozzle can enhance the performance of high-pressure cleaning [19]. Peng analytically investigated the effect of the spray distance of the nozzle on the cleaning performance [20], and Yang studied the effect of the installation of nozzle angle on the spray pressure in the spray system [21]. A theoretical model for evaluating the optimal and critical distances for cleaning using high-pressure water was proposed by Guha [22]. Xu and Chen pioneered the studies on spray properties and their effect on nozzles at high pressures [23, 24]. Medan optimized a cleaning equipment to measure the reaction force of the spray nozzle and to determine the main factor that affects high-pressure cleaning [25]. Zhong studied surface erosion according to nozzle type [26], and Sertore examined the state of a nozzle based on the injection force measured by a load cell [27]. However, these research studies employed very high pressure for special purposes; the findings are therefore not applicable in cleaning the outer walls of buildings. This is because high pressure has certain limitations in terms of surface damage, robot orientation control, and commercialization.

To overcome these limitations, we propose a method for cleaning outer walls of buildings using high-pressure water. Unlike the conventional robot or the previously mentioned very-high-pressure method, the proposed device uses high-pressure water. It can clean over a wider range by varying the injection angle and direction of the nozzle. The robot can clean windows continuously while overcoming obstacles because this method does not use cleaning device that needs to be in direct contact with the wall for the cleaning. In addition, it does not generate uncleaned zones because the spray direction is variable. It is also possible to clean areas that are difficult to access using a brush.

The quantitative measurement of cleaning performance is also an important issue in this study. Various measurement methods have been introduced by researchers to evaluate the cleaning performance of devices [28, 29]. Kang evaluated contamination by using infrared transmissivity [30], and Moon et al. used an experimental method to evaluate cleaning performance [31].

A cleaning device that uses high-pressure water for cleaning is introduced in this study. A method to determine the optimal design parameter conditions and ensure reliable cleaning

performance is also described herein. Design parameters that influence the cleaning performance of the proposed cleaning devices were selected.

The selected performance design parameters were optimized experimentally to maximize the cleaning performance. The experimental results of cleaning by high-pressure water were evaluated according to image data obtained using a digital camera. We adopted Taguchi orthogonal array for effective experimentation [32, 33]. The Taguchi method enables the optimization condition of experiment factors to realize inexpensive and rapid product design. Further, it is possible to determine more effective conditions and verify these (and thereby, improve product performance) with an identical number of experiments. The high-pressure water that the robot uses for cleaning generates reaction and impact forces. In this study, to examine the influence of the reaction force on the propeller thrust and that of the impact force on the surface of the building's exterior wall, the magnitudes of the reaction force and the impact force were measured at the nozzle and on specimen, respectively.

The remainder of this paper is organized as follows. Section 2 introduces the overall cleaning system and working environment. Section 3 presents the configuration of the proposed cleaning device. Section 4 defines the optimization problem and experimental design, including the objective function, design parameters, and user condition. Section 5 describes the results of an experiment performed with a test bench and presents a discussion. Finally, the concluding remarks are presented in Section 6.

## 2. Overall description of cleaning robot and devices

A robot installed in a high-pressure cleaning module is shown in Fig 2. The robot can move over the building's surface using two ropes that are secured at the top of the building. Additional devices are installed on both edges of the building's top to secure the ropes' ends. This system involves simple installation and enables the operation of robots for buildings without the use of gondolas or separate winch devices. The robot uses two winches that are embedded into it. The force for attaching on to the wall is provided by the propeller. This type of robot system is relatively insensitive to the reaction force against the spray pressure and the force balance of the nozzles' pressure. Therefore, the operational stability is higher than that of a robot that uses a suction plate. In addition, severe issues such as the occurrence of simultaneous suction of water and air, the required payload for the devices, and damages to the adsorption plate caused by foreign matter and wear do not arise. In total layout of the robot, it is advantageous on a space efficiency because of the simple structure of the cleaning unit.

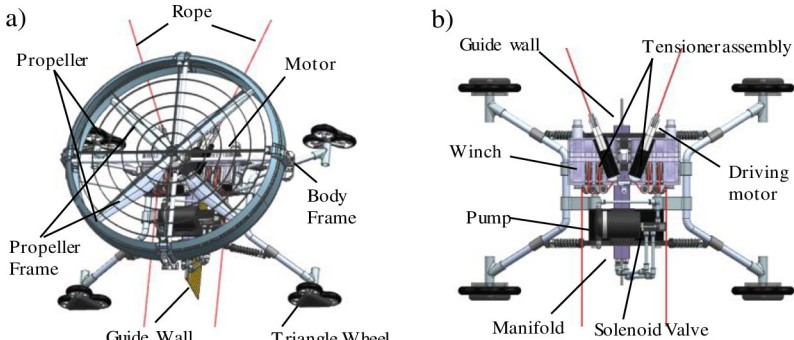

**Fig 2. Configuration of winch-mounted wall-climbing robot [39, 40].** The robot largely consists of three parts: a propeller thrust unit, winch unit, and cleaning and frame unit. The cleaning unit is placed at the bottom of the robot considering water leakage. a) Overview of winch-mounted wall-climbing robot (b) Assembly of the winch unit and cleaning and frame unit.

Triangular wheel sets such as star wheel of MSRox [34] (each wheel with a diameter of 150 mm) are applied for traversing on walls. These can overcome obstacles with heights less than 100 mm, such as window frames. To prevent damage to the building's surface by slippage, wheels made of a soft material were applied. In addition, a suspension device and air-injected wheels were adopted to minimize the transmission of impact force and to protect the surface. Unlike conventional robots, this robot sprays high-pressure water continuously while overcoming obstacles. This is an important advantage for reducing the amount of uncleaned zone compared with that for existing cleaning robots.

The cleaning device can be assembled in parts or as a single device. Here, the device was assembled in two parts considering the space efficiency of the robot. For a large weight (2 kg), the pump should be placed close to the line of gravity of the robot to enhance control efficiency and thereby, minimize the influence of the moment of inertia. In addition, the cleaning unit components were placed at the bottom of the robot because of concerns of water leakage. A guide wall was installed to prevent recontamination of the cleaned area. The cleaning range can be altered by varying the angle of the replacement nozzles, which is a significant advantage with regard to mass production and robot maintenance.

To ensure effective cleaning, the robot control algorithm optimization should consider various scenarios, e.g., spray control for continuous cleaning while overcoming obstacles, movement control against the reaction force generated by the high-pressure water spray, and orientation stability control at the beginning of spray. Therefore, the capability for cleaning and for overcoming obstacles must be evaluated simultaneously. It would be more effective to test after achieving control optimization. Therefore, the optimization of the entire system would be addressed in future studies. The optimization of the cleaning performance, which is the main focus of this study, is critical for the overall system. If the cleaning performance is not satisfactory, it would be ineffective to apply the robot at cleaning sites. Therefore, it should be designed to achieve a high level of cleaning performance. In the following sections, the specifications for the design and optimization of the cleaning performance are presented in detail.

## 3. Specification of proposed cleaning device

The proposed cleaning device consists of nozzles for spraying high-pressure water, a pump for generating high pressure, a pressure sensor for measuring the inlet pressure of the nozzle unit, and a wall for preventing recontamination. Four nozzles are used to create a spray zone with minimal overlap. A manifold is adopted to minimize the pressure loss in each nozzle and maintain a uniform spray pressure.

The criteria for selecting the nozzle tip were based on two aspects: 1) the impact efficiency of spray water, which represents the capability for removing contaminants, and 2) the uniformity of the shape of the water spray at the nozzle tip, which determines the uniformity of cleaning performance. According to the data sheets provided by the manufacturer, the impact efficiency of the flat-type nozzle at 50° is 10%, whereas those of the full-cone and hollow-cone-type nozzles are < 1% and 1%, respectively [35, 36]. The spray shape uniformity according to the nozzle type was verified at the designated pressure. The flat-type nozzle displays higher spray uniformity with a fan shape and is the most widely used type of nozzle for cleaning. In addition, it displays a higher impact efficiency than that of the hollow-cone- or full-cone-type nozzle. Finally, the HM_V type (Hanmi Nozzle co. ltd [35]) of flat nozzle was selected, as shown in Fig 3. A single unit nozzle made of stainless steel was selected to ensure reliability.

The nozzles sprinkle a specialized solution that contains an alcohol-based material to clean organic pollutants. Pulse width modulation (PWM) flow control and a solenoid valve are used to control the amount of cleaning solution applied. The water pump can discharge a maximum

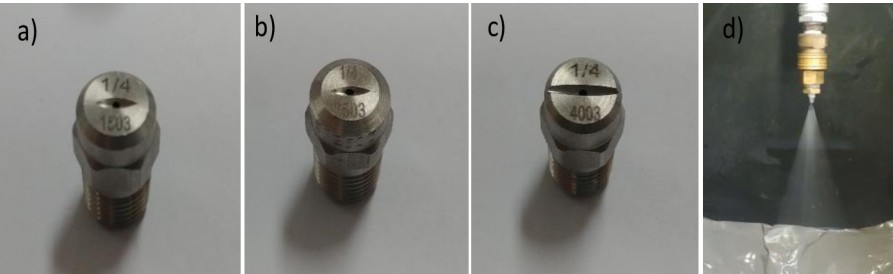

**Fig 3. A flat-type nozzle displays good spray uniformity.** In terms of impact efficiency, it displays higher cleaning performance than those of hollow-cone- or full-cone-type of nozzle. (a)–(c) are flat nozzle tips. Their spray angles are 15˚, 25˚, and 40˚ respectively, (d) spray angle of 40˚ at 6 bar [35, 36].

pressure of 10 bar. However, to ensure working stability, the applied pressure is designed to be < 80% of the maximum discharge pressure. The discharge pressure is controlled by the rotational speed of the pump. A pressure sensor is installed at the inlet of the nozzle to monitor the pressure supplied to the nozzle. It enables the accurate control of the pressure supplied to the nozzle. The pressure sensor detects the status of the pump when the water supply is intermittent or the pump does not operate. During cleaning, the sensor monitors the pressure drop of the pump. In addition, it detects the variation from the designated injection pressure when the device is overcoming a low obstacle. The measured data from the sensor is transmitted to the controller to maintain the pre-set pressure.

A guide-wall is important to prevent recontamination of the cleaned area, although this is not directly related to the cleaning performance. It is installed between the manifold. We designed the nozzle spraying direction to be tilted at an angle of approximately 10˚ toward the transfer direction, to minimize the splashing of the contaminant toward the cleaned area. Although the fundamental specification ranges of the cleaning appliance have been selected, the detailed design parameters must be verified experimentally as described in the following section.

## 4. Planning for robust optimal design and experimental setup

### 4.1. Control parameters and user condition

We examined the sensitivity of the performance of the high-pressure water cleaning to the following variables: nozzle inlet pressure, nozzle spray distance, and spray angle. The variables that are to be optimized to achieve the maximum performance are presented in Table 1 and are shown schematically in Fig 4. The nozzle inlet pressure is related to the flow rate as follows [21]:

$$\frac{Q_1}{Q_2} = \left(\frac{P_1}{P_2}\right)^n \tag{1}$$

**Table 1. Control parameters and user condition for optimization of the cleaning device.**

| | Parameter | | Level 1 | Level 2 | Level 3 |
|---|---|---|---|---|---|
| *Control Parameters* | Nozzle inlet pressure (bar) | A | 3 | 6 | 8 |
| | Spray distance (mm) | B | 200 | 300 | 400 |
| | Spray angle (˚) | C | 15 | 25 | 40 |
| *User condition* | Cleaning speed (m/s) | D | 3 | 6 | - |

Here, Q, P, and n represent the flow rate, pressure, and specific gravity, respectively, of the fluid. According to Eq (1), we control the pressure rather than the flow rate, as shown in Table 1. Considering the large impact force and the most frequently used nozzle types at cleaning sites, three spray angles of flat nozzle were selected. Considering the cleaning performance, the impact force is reduced considerably at over 40˚, and satisfactory cleaning performance is unlikely. The spray distance is defined as the distance between the building surface and nozzle tip. The distance is closely related to the overall layout of the cleaning robot. The space between the outer wall and nozzle tip as well as the cleaned area are closely related to the robot's structure. In addition, the distance is a factor that determines the maximum height of obstacles that can be overcome. It was determined to be 200–400 mm considering the performance and the robot's layout.

The cleaning speed is a user condition that determines user operating speeds. Therefore, two levels of descending speed of the robot (3 and 6 m/min) were selected to verify the best and worst parametric combinations, respectively, as shown in Table 1 and Fig 4.

## 4.2. Evaluation of cleaning performance

All the parameters should be optimized to maximize the cleaning performance. Therefore, it is important to design and perform an evaluation of the cleaning performance. Although many researchers have recommended methods for measuring and evaluating the contamination on the exterior walls of buildings, there are no international or domestic standard. In addition, there is no standard definition of the cleanliness of exterior walls. The perspectives differ among individuals and industries. Therefore, an image processing method [29, 38] involving the acquisition of photographs before and after cleaning was used for a quantitative evaluation of the cleaning performance in this study.

Fig 5 shows the setup for evaluating the cleaning performance, as well as the result sample. Photographs were captured before and after cleaning by using the same camera, which was set at a constant height and with constant illumination using an aluminum frame. Thus, the post-cleaning data were obtained (as shown in Fig 5C), and the cleaning performance was expressed as the ratio of the area of contamination before cleaning to that after cleaning, after the two photographs were filtered to a certain threshold value. Freeware was used to evaluate the area on which dust remained after the test [38].

## 4.3. Objective function and experimental design

The selected design parameters are large-the-better characteristic for the cleaning performance. Because a higher cleaning performance would be obtained when a wider area is

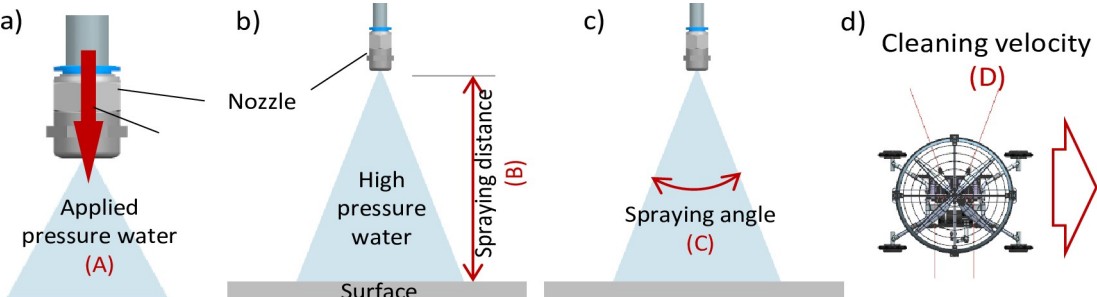

**Fig 4.** Control parameters (a–c) and a user condition (d). a) Nozzle inlet pressure, b) Spray distance, c) Spray angle. d) Cleaning speed of the robot [29, 37].

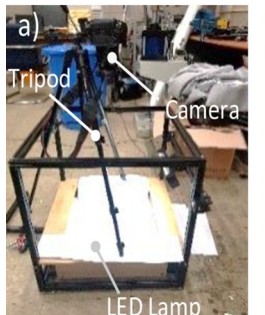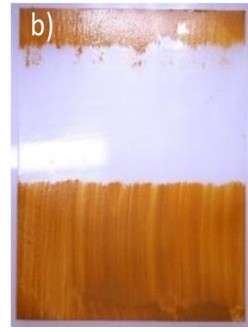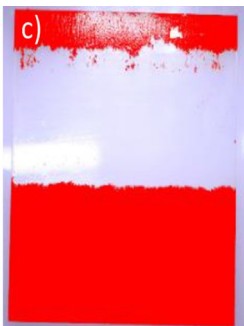

**Fig 5. Device setup for the image data measurement and conversion to image data from the test results.** a) Device setup to capture photographs for the image data, b) A test result photograph after cleaning, c) Image data converted from the photographs [29, 38].

cleaned, the signal-to-noise ratio (SNR) was applied as shown in Eq (2):

$$SNR = -10log_{10}\left|\frac{\left(\frac{1}{y_1}\right)^2 + \left(\frac{1}{y_2}\right)^2 + \cdots + \left(\frac{1}{y_n}\right)^2}{n}\right| \qquad (2)$$

Here, $y_i$ represents the measurement data, and n represents the number of data. Taguchi orthogonal matrices [32, 33] provide a highly popular method to design and perform experiments using standardized arrays. These special orthogonal arrays define the minimum number of experiments that could yield the sensitivity of all the factors that affect the performance. We decided to use the $L_9(3^3)$ of the orthogonal array on the basis of the three levels of design variables. The experimental sequence and the combination of variables are presented in Table 2. Two levels of user conditions (device cleaning speeds of 3 and 6 m/min) were considered, as shown in Fig 4.

## 4.4. Test bench configuration and experimental setup

The test bench was designed with two parts: a water spray unit and a cleaning-performance evaluation (Fig 6). A water spray unit consists of a load cell, a pressure sensor, and a nozzle device assembly. In addition, a cleaning-performance evaluation unit has a window frame with a load cell and transfer devices with two axes. In the water spray unit, the nozzle feed pressure was the control factor. The pressure was varied by modifying the rotational speed of the pump. In addition, a pressure sensor monitored the nozzle inlet pressure. Although the nozzles

**Table 2. $L_9(3^3)$ orthogonal array and evaluation results of image data.**

| Exp.# | Target function variables | | | Descending speed = 3 m/min | | | Descending speed = 6 m/min | | | SNR |
|---|---|---|---|---|---|---|---|---|---|---|
| | *A* | *B* | *C* | *Panel 1* | *Panel 2* | *Panel 3* | *Panel 1* | *Panel 2* | *Panel 3* | *(dB)* |
| | *(Level)* | *(Level)* | *(Level)* | *(%)* | *(%)* | *(%)* | *(%)* | *(%)* | *(%)* | |
| *1* | 1 | 1 | 1 | 12.89 | 10.53 | 10.11 | 0.13 | 0.26 | 0.14 | -12.98 |
| *2* | 1 | 2 | 3 | 52.30 | 43.93 | 40.41 | 27.82 | 41.84 | 37.92 | 31.70 |
| *3* | 1 | 3 | 2 | 31.22 | 28.67 | 24.01 | 9.75 | 1.79 | 7.95 | 12.44 |
| *4* | 2 | 1 | 3 | 36.20 | 32.72 | 41.29 | 10.66 | 19.85 | 5.28 | 20.83 |
| *5* | 2 | 2 | 2 | 28.50 | 21.16 | 31.46 | 2.41 | 1.87 | 5.85 | 10.87 |
| *6* | 2 | 3 | 1 | 1.18 | 1.98 | 0.67 | 0.04 | 0.92 | 2.43 | -19.00 |
| *7* | 3 | 1 | 2 | 72.80 | 74.11 | 72.60 | 23.55 | 45.47 | 43.60 | 32.50 |
| *8* | 3 | 2 | 1 | 0.85 | 2.17 | 2.75 | 0.21 | 2.73 | 1.94 | -6.12 |
| *9* | 3 | 3 | 3 | 98.12 | 70.58 | 91.23 | 4.00 | 28.48 | 48.79 | 19.69 |

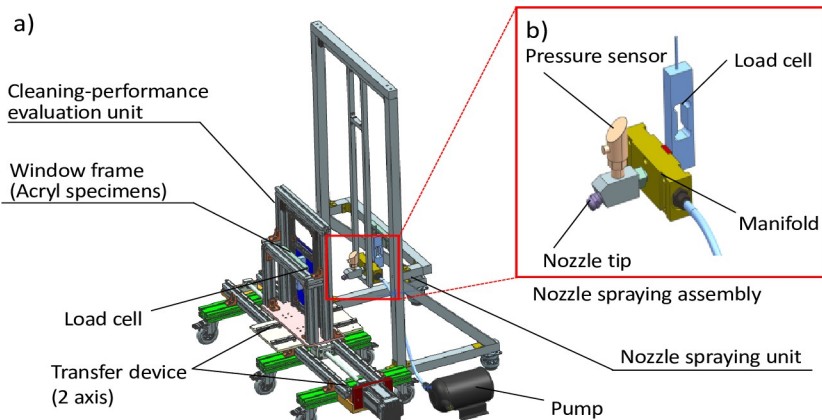

**Fig 6. The test bench consists of two parts: A cleaning performance evaluation unit for investigating spray characteristics and a nozzle spray unit for implementing movement of the robot for cleaning.** Each unit has a load cell to measure the reaction force and impact force. (a) Overall schematic of test bench and that of (b) the water spray unit.

used in the experiment had identical orifice shape, their spray angles were different [35]. The cleaning-performance evaluation unit was designed with a distance-adjustment device for evaluating the cleaning performance according to the distance between the nozzle and panel. In addition, the speed of motion of the panel was considered as a user condition. Two load cells were applied: One was used to measure the reaction force from the high pressure on the nozzle inlet, and the other was located on the back of the window panel and was used to identify glass damage.

One of the important factors affecting the cleaning performance is the pressure. The rotational speed of the pump was adjusted to three values (as shown in Table 2), and a pressure sensor was installed in the nozzle inlet to measure the supplied pressure. A tube with an outer diameter of 12 mm and specially manufactured nipples were used to minimize the pressure drop of the pipeline. In addition, manifolds were employed to enable convenient nozzle-tip replacement, achieve a uniform pressure, and minimize the influence of the pressure-sensor mounting, as shown in Fig 7.

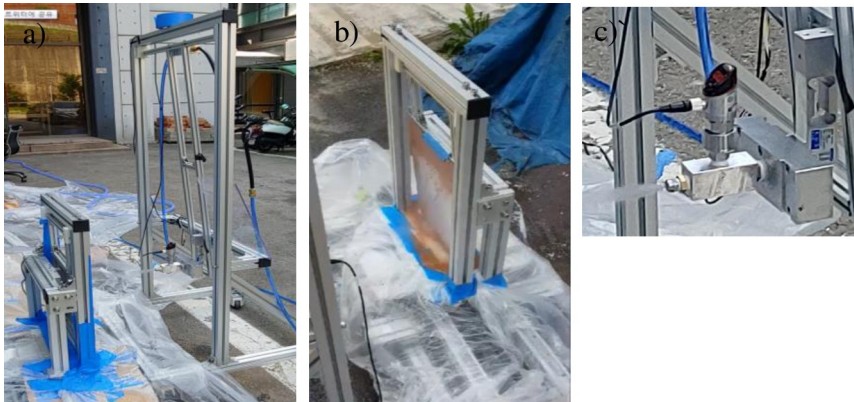

**Fig 7. Experimental test bench setup for evaluating the cleaning performance.** a) Test scenario with high-pressure water cleaning, b) Cleaning performance evaluation unit and transfer devices in two directions (X, Y). c) Water-spray-unit assembly. Specifications of main parts: Pump: 8905-902-290(SHURflo), Pressure sensor: GP-M025, (KENYCE), Load cell: BCA_5(CAS), Nozzle: Flat type, HM_V5 (Hanmi Nozzle.co.Ltd) [35].

## 5. Test results and discussion

### 5.1. Sensitivity analysis and results review

The orthogonal array and the experimental results are summarized in Table 2. As shown on the right part of the table, to obtain the image data result for each plate, the area cleaned by the high-pressure water was calculated by comparing the window surfaces before and after cleaning. The SNR was calculated using the experimental data via Eq (2). As indicated by the test results in Table 2, the cleaning performance achieved with a high pressure was better than that achieved with a low pressure, and the cleaning performance at a low descending speed was better than that at a high descending speed. However, the robot's descending speed was not more sensitive than nozzle inlet pressure, spray distance, and spray angle, as shown in the experimental results in Fig 9G–9L.

The result of the sensitivity analysis of the selected design variables is shown in Fig 8. This is calculated via Eq (2). The optimal combination of the design variables was determined as follows: nozzle inlet pressure = 8 bar, spray distance = 0.2 m, spray angle = 40˚, and transfer speed of the specimen = 3 m/min. The highest and lowest sensitivities were to the nozzle inlet pressure and spray angle, respectively. The cleaning performance increased as the pressure increased. In addition, the spray distance exhibited an inverse relationship with the cleaning performance.

It is noteworthy that the cleaning performance decreased significantly below a critical spray angle. As shown in the experimental results in Fig 9, although both ends of the spray area are clean, the results are not effective in the central part. We conjecture that this is because the flow rate was not uniform over the water-spray area when the spray angle of the nozzle was 15˚ or 25˚. In particular, an identical pattern of cleaned areas appeared on the specimen with these small spray angles, as shown in Fig 9D–9F. These results of the cleaning pattern caused by the flow imbalance at small angles are highly helpful and provide guidance for developing devices for cleaning external walls using a high-pressure spray that display good performance.

The experimental results shown in Figs 8 and 9 reveal that a higher cleaning performance was achieved with the condition characterized by a high nozzle inlet pressure, long spray distance, and wide spray angle than with that characterized by a small nozzle inlet pressure, short spray distance, and narrow spray angle. This is because a wide spray angle can cover a wide range of cleaning area with a high nozzle inlet pressure. Within the experimental set of spray distances, the cleaning performance was more sensitive to the injection pressure than to the

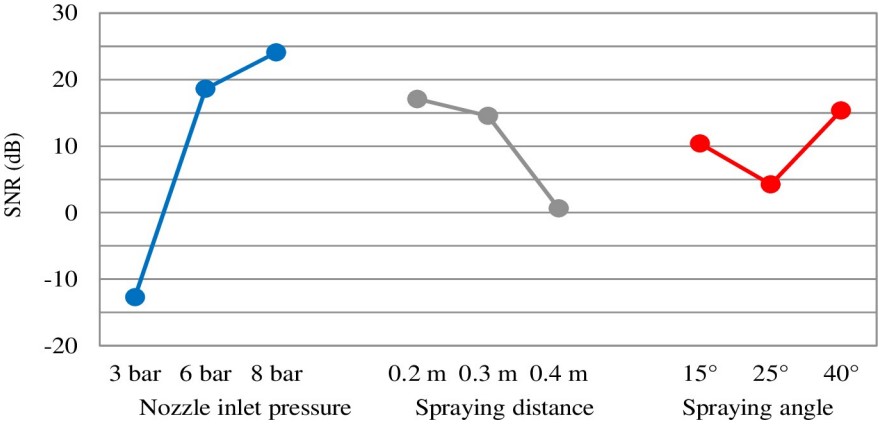

**Fig 8. Sensitivity analysis results of the control parameters based on Eq (2).**

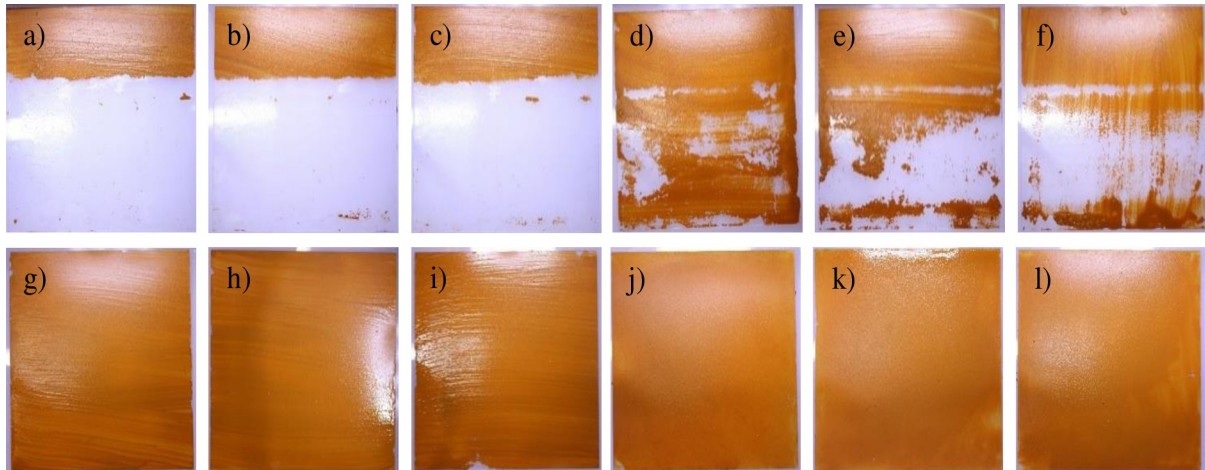

**Fig 9. Test results for the highest and lowest cleaning performances.** The highest performance results were those of Exp.#7 (Table 2) for the user condition of 3 m/min, as shown in a)–c). d)–f) display the results for 6 m/min. The worst performance result was for Exp. #6 under both the user conditions (3 mm/s and 6 mm/s).

spray distance, as shown in Fig 8. Although the cleaning performance under the condition characterized by a high nozzle inlet pressure and narrow injection angle is normally good, it is not better in terms of the definition of cleaning performance. This is because the total area to be cleaned by the robot per unit distance moved is marginal.

## 5.2. Verification experiment and discussion

An experiment for verifying the optimal values of each factor was performed based on the results of the sensitivity analysis. The optimum conditions were as follows: nozzle input pressure = 8 bar, spray distance of the nozzle = 200 mm, and injection angle = 40˚. The pressure could not exceed 8 bar owing to the pump's limitation, as mentioned earlier. The injection angle and spray distance were determined according to the layout of the robot and the sensitivity analysis results. The results of the verification experiment based on the optimum conditions were better than those of Experiment #7, as shown in Fig 10.

The reaction force at the nozzle and the impact force at the acryl specimen when the nozzle injected high-pressure water for cleaning were measured. This was performed to investigate 1)

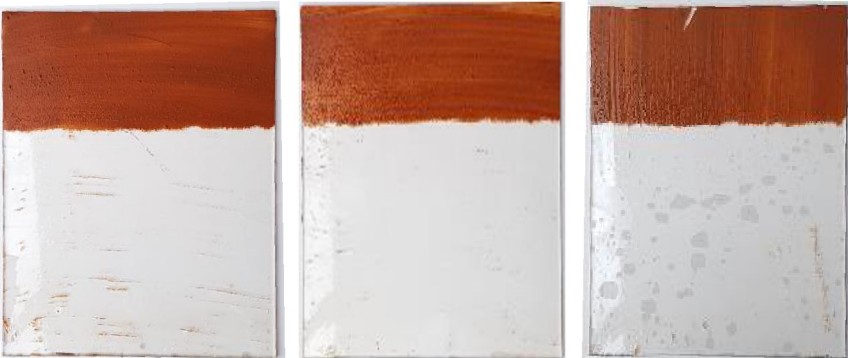

**Fig 10. The results of verification test performed under the optimal conditions.** Nozzle inlet pressure: 8 bar; spray distance: 0.2 m; injection angle: 40˚. Image data results: 74.8% (average).

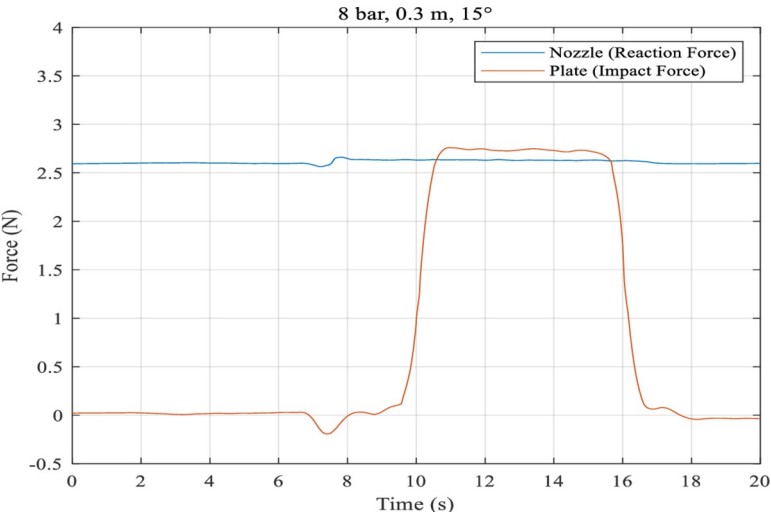

**Fig 11. Reaction force on nozzle and impact force on panel under the condition of 8 bar, 0.3 m, 15˚.**

the influence of the reaction to the thrust force from the propeller that is applied for the robot to adhere to the wall and 2) the damage to the building's outer wall by the impact force. As shown in Fig 11, the reaction force at the nozzle and the impact force on the outer wall were approximately 2.6 N and 2.7 N, respectively. The nozzle spray reaction force was approximately 4.5% of the thrust force of the propellers. This indicates that the spraying at a pressure of 8 bar did not significantly affect the absorption or damage the exterior wall of the building.

## 6. Conclusion and future works

In this study, a method of spraying high-pressure water for cleaning the outer walls of buildings was developed, and the cleaning performance was optimized. The proposed cleaning device was installed on a robot equipped with a winch. Cleaning was performed by spraying high-pressure water using nozzles, a high-pressure pump, and a guide wall to prevent recontamination. The control parameters were optimized by utilizing the Taguchi optimization method for different descending speeds. The design variables were identified based on experience to evaluate the cleaning efficiency through image processing. We carried out a sensitivity analysis to identify the design variables to which the cleaning performance is most sensitive. The following are the variables arranged in decreasing order of influence on the cleaning performance: nozzle inlet pressure, spray distance, and injection angle. The cleaning performance was not good for a slow speed of motion under the condition of a small spray angle or low spray pressure. We verified the optimized conditions through a test under the following conditions: nozzle inlet pressure = 8 bar, spray distance = 0.2 m, nozzle injection angle = 40˚, and descending speed = 3 m/min. The reaction force and impact force generated by the pressure were 2.6 N and 2.7 N, respectively. These are not adequate to affect the propeller thrust force or damage the building surface.

In the near future, based on the results of the proposed method, lab tests are expected to be carried out with the upgraded ascender control algorithm for overcoming obstacles. Through this test, the cleaning quality of the high-pressure-water spray while overcoming obstacles would be verified, and the uncleaned zone generated by obstacles would be assessed. Thus, a study will be conducted to reduce the uncleaned zone and thereby achieve good cleaning quality. Subsequently, a field test will be conducted to verify the cleaning performance of the proposed cleaning method.

## Author Contributions

**Conceptualization:** Youngjoo Lee, Daesung Kwon, Changmin Park, Myoungjae Seo, TaeWon Seo.

**Data curation:** Youngjoo Lee, Daesung Kwon, Changmin Park, Myoungjae Seo.

**Funding acquisition:** TaeWon Seo.

**Investigation:** Changmin Park, Myoungjae Seo.

**Writing – original draft:** Youngjoo Lee, Daesung Kwon.

**Writing – review & editing:** Youngjoo Lee, Daesung Kwon, TaeWon Seo.

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
