## [Decision Letter · Decision Letter 0]

28 Aug 2020

PONE-D-20-12174

Parametric study on the automated high-pressure water-based window cleaning device

PLOS ONE

Dear Dr. Seo,

Thank you for submitting your manuscript to PLOS ONE. After careful consideration, we feel that it has merit but does not fully meet PLOS ONE’s publication criteria as it currently stands. Therefore, we invite you to submit a revised version of the manuscript that addresses the points raised during the review process.

We look forward to receiving your revised manuscript.

Kind regards,

Hongbing Ding, Ph.D.

Academic Editor

PLOS ONE

Journal Requirements:

2. Please provide vendor details of all equipment and materials used. Please ensure that the methods section is described in enough detail for another researcher to reproduce the findings.

3. Please clarify where the data underlying the findings of the study can be found. We note for instance that no Supporting Information has been provided.

4.We suggest you thoroughly copyedit your manuscript for language usage, spelling, and grammar. If you do not know anyone who can help you do this, you may wish to consider employing a professional scientific editing service.  

Additional Editor Comments (if provided):

Thank you for submitting your manuscript to PLOS ONE. The reviewers recommend reconsideration of your paper following major revision. I invite you to resubmit your manuscript after addressing all reviewer comments.

Reviewers' comments:

Reviewer's Responses to Questions

**Comments to the Author**

1. Is the manuscript technically sound, and do the data support the conclusions?

Reviewer #1: Partly

Reviewer #2: Yes

2. Has the statistical analysis been performed appropriately and rigorously? 

Reviewer #1: Yes

Reviewer #2: N/A

3. Have the authors made all data underlying the findings in their manuscript fully available?

Reviewer #1: Yes

Reviewer #2: Yes

4. Is the manuscript presented in an intelligible fashion and written in standard English?

Reviewer #1: No

Reviewer #2: No

5. Review Comments to the Author

Reviewer #1: Summary

The authors present a technique for window cleaning of high-rise buildings---an increasingly prominent problem, as many high-rise buildings have been constructed recently. Manual window cleaning of such structures is dangerous for various reasons. Various robots have been proposed to assist with this problem. In this paper, the authors propose a method of spraying high-pressure water by using a pump and nozzle. Unlike other approaches, utensils, e.g., brush, squeeze, manipulator, typically used for manual cleaning, are not used in the proposed plan. Omitting such tools has essential implications for maneuvering around obstacles, such as building facades. The method was evaluated by polluting windows, using the robot on the windows, and quantitatively comparing before and after images. A field study is planned for the future.

Pros

- Relevant and timely topic. Cleaning windows on high-rise buildings manually is dangerous.

- The proposed approach seems to have substantial advantages over existing methods. The benefits are particularly significant when considering that buildings may have facades and other obstacles.

- Preliminary results are encouraging.

Cons

- The quality of the writing is poor. Not even spell check was seemingly executed on this article.

- No field study. The current research may not be complete.

- The novelty of the proposed approach over existing approaches is not apparent.

Major Details

- Why do existing solutions not satisfy you? What is wrong with them, specifically? What are the disadvantages of those approaches?

- How about qualitative analysis of performance?

- What are the effects of wheel damage on the buildings that your device may cause?

- "In case of higher than 100mm of obstacles, a separate mechanism is needed." -> Isn't this the problem you are solving over existing approaches, i.e., not being hindered by obstacles without using separate mechanisms?

- "In other words, the further away the distance, the poorer the cleaning performance." -> Isn't this expected?

- Why weren't obstacles used in the evaluation?

- Why does slow speed impede performance? Isn't this counterintuitive?

Minor Details

- Title: The paper is not really about a study. Instead, an approach is proposed; the study helps evaluate the proposal. I suggest changing the title to something like: "An Automated Technique for High-pressure Water-based Window Cleaning and Accompanying Parametric Study" (the latter half about the study may be dropped).

- The motivation is that many high-rise buildings need cleaning. However, with COVID-19, many of these buildings are empty. COVID-19 may have a significant impact on commercial real estate. How does this affect the impact of the proposed approach? Are there other contexts where the device can be used in light of COVID-19?

- The abstract is too long—too many details.

- It would be better to have the figures at the top of the page (more comfortable to read).

- The conclusion presents new information not found in the introduction (recontamination). The opening and conclusion should mirror each other. If this information is essential, it should also be included in the introduction. Otherwise, it should only be in the discussion section of the experimentation.

- No future work section. Highlighting future work is crucial since you are planning future studies.

Reviewer #2: This paper involved a method to spray high pressure water in the process of cleaning exterior building walls, in addition to optimizing cleaning performance. Although the premise is clear, the outcome is significant for an effective performance of building exterior wall cleaning and maintenance, and the developed methods and data are relatively sound, the paper is extremely hard to read, given the poor English writing, improper sentence structure, multiple typos and grammatical errors, making it not only unsuitable for submission in the current state, but also ambiguous to read and understand technically, which often does not allow the reader to grasp some vital information to the understanding of the methods, procedures, and results. The paper cannot be submitted in this state of poor writing and requires thorough revision by a native English speaker. Further comments might even arise upon a re-read once some of the concepts and procedures become more clear.

Although the research problem identified and discussed in section 1 is understood, the authors are encouraged to elaborate more on the problem formulation, with clear justification of the need for their research. The authors mention some relevant studies, but do not go in length in delineating how past studies and precedents have failed to achieve the desired objective regarding exterior wall cleaning performance (more in terms of metrics and data, rather than anecdotal information). This should help ground more the research and justify the discussed methods and procedures later on. Perhaps a brief discussion about how precedent studies specifically addressed similar methods and cleaning parameters and other relevant data could help bring more justification to decisions such as the choice of the Taguchi method and the image evaluation method (which are currently described very briefly and there is no elaborate grounding of why they are selected for this specific experiment and study).

Section 2 is mostly ambiguous and it is not clear from the current version of the paper whether the discussed topics related to cleaning devices and equipment are mostly guidelines to follow, methods adopted by the authors, or work developed by the authors. This needs to be classified and demonstrated clearly. Are the cleaning devices mentioned used by the authors? Or are they just referred to as examples of previous work? This is not clear, perhaps more due to the ambiguous language more than anything else. There are also some strong assumptions in the third paragraph in page 3 that require some clarification. If some of the cleaning devices and robots are to be adopted, there needs to be some more explanation regarding specific relations to the proposed methods described later on and how they fit together. This is missing.

Section 3 lays out well some of the specifications for the proposed device, but requires some more explanation, especially in the second paragraph discussing the selection criteria. There appear to be some gaps in the description of the selection criteria. This requires further elaboration.

Most of the figures, especially 4, 5, 9, and 10 are unreadable and difficult to read. These are crucial illustrations that should be much more developed and more clear, perhaps enlarged in size, to demonstrate the suggested differences in cleaning performance, and so they require much more clear images and image resolution. Some figures are misplaced, such as figure 6, which needs to belong under section 4.4. Table captions need to be above the tables not below.

The first two paragraphs in page 5 are relatively confusing in relation to the tables and figure that follow. Please describe very clearly the content of each table (in a more elaborate discussion) so that the flow of information reads well. There is no adequate discussion regarding the mentioned levels (1, 2 and 3) and how they relate each of the parameters. Table 2 and 3 are quite short and their inclusion is questioned. Perhaps another tabulation format can be included, otherwise they could be eliminated and included as text description, but with further explanation.

Section 4.2 lacks both adequate description of how previous research fall short in terms of method or standards (this requires specific citations and identification of gaps, and not just general talk), and a sufficient description of process that is conducted by the authors (this is quite truncated and not described clearly). The authors do not mention also the "certain threshold value" for cleaning in the first paragraph in page 6, adding to the ambiguity in this section. Section 4.3 is quite hard to read and understand. The only useful piece of information is the equation which is utilized effectively later on in the results. Other than that, the text needs substantial work to be understood in the first place. The conclusion of the paragraph following the equation needs to be clear. The experiment setup in section 4.4 lacks organization and clarity, making it also hard to follow. Please organize the description of the setup into clear chunks of information that are easy to follow. Table 4 is primarily related to the evaluation results, so it seems more appropriate to locate this table later on in section 5, rather than under 4.4 which is more related to the methods and procedures. You come back to it later and recall it in the results section in a confusing manner, so it suits that location in the text better. This should organize the flow of ideas better.

The sensitivity analysis described in section 5 is interesting and revealing of your findings. However, there seem to be some anecdotal observations and assumptions in the third paragraph in page 8. These should be avoided; please stick to the data and pure results. The interpretation of the results should be addressed far more scientifically than what is mentioned here. The results also need more elaboration in the second paragraph in page 8; seems quite short and truncated. In other words, how you arrive at the optimum results (0.2m spraying distance, 40 degrees angle, and 3m/min transfer speed) is the core of your findings. This needs to be analyzed, interpreted and discussed widely. The confirmation section (5.2) reads well but needs more elaboration. Figures need to be more visually clear. There needs to be a discussion here or in the earlier section about comparisons to base cases.

The paper ends relatively in an anti-climactic manner. There needs to be an elaborate discussion section that goes thoroughly through the interpretation of these results and how they relate to the robotic cleaning device and the consequences of the optimized values. There also needs to be a macro discussion that relates to buildings, and the generalizability of such findings. Are these results applicable for example and true for all locations, conditions, heights? Many other factors need to be considered, such as the climate, dust, etc. It is understood that this is not the scope of the paper, but there needs to be a clear description of the assumptions of the paper, in terms of weather conditions, building orientation, wind speed, etc. that all might critically affect some of the results/parameters such as spray angle and spraying distance. What is deduced here in this paper might apply for example in a southern facade but not in a western facade, and at certain elevation levels not others, and in specific climatic zones and not others. Grounding the findings of your research should specify in detail these assumptions or at least the conditions under which you conducted your experiments. If this is in a controlled environment, what must be accounted for in field testing? There also needs to be a discussion and justification regarding the effect of the suggested 2.6-2.7N impact force and how the assessment of minimal damage to the building surface is justified; this is not clear. There needs to be more elaboration regarding how to carry your findings from the experimental setup into field tests; at least in terms of the assumptions in each case and what additional experiments need to be conducted to transfer to real conditions and what to account for.

6. PLOS authors have the option to publish the peer review history of their article (what does this mean?). If published, this will include your full peer review and any attached files.

Reviewer #1: No

Reviewer #2: No

---

## [Author Response · Author response to Decision Letter 0]

27 Sep 2020

Thank you for the valuable comments on this article. We carefully revised the paper according to the comments. The main corrections are as follows:

The title, abstract and introduction parts were revised to clearly point out our novelty and contributions of the proposed method.

All Figures in the text were revised image the clarity to increase the readability for the paper.

Tables containing variables are combined into one table to help easy understand variables and pictures.

To identify the cleaning performance, a simple additional test was performed, and the contents were added in the answer sheet.

To improve the readability, sentences were revised grammatically and unnecessary words with no meaning were deleted for easy reading. (The editing is done by a professional editor (https://www.editage.co.kr/_

Please refer the attached revision summary and revised draft for details.

---

## [Decision Letter · Decision Letter 1]

3 Nov 2020

Automated technique for high-pressure water-based window cleaning and accompanying parametric study

PONE-D-20-12174R1

Dear Dr. Seo,

We’re pleased to inform you that your manuscript has been judged scientifically suitable for publication and will be formally accepted for publication once it meets all outstanding technical requirements.

Kind regards,

Hongbing Ding, Ph.D.

Academic Editor

PLOS ONE

Additional Editor Comments (optional):

The authors have done a good job in revising the manuscript. Now it can be accepted for publication in PLOS ONE.

Reviewers' comments:

Reviewer's Responses to Questions

**Comments to the Author**

1. If the authors have adequately addressed your comments raised in a previous round of review and you feel that this manuscript is now acceptable for publication, you may indicate that here to bypass the “Comments to the Author” section, enter your conflict of interest statement in the “Confidential to Editor” section, and submit your "Accept" recommendation.

Reviewer #1: All comments have been addressed

Reviewer #2: All comments have been addressed

2. Is the manuscript technically sound, and do the data support the conclusions?

Reviewer #1: Yes

Reviewer #2: Yes

3. Has the statistical analysis been performed appropriately and rigorously? 

Reviewer #1: Yes

Reviewer #2: Yes

4. Have the authors made all data underlying the findings in their manuscript fully available?

Reviewer #1: Yes

Reviewer #2: Yes

5. Is the manuscript presented in an intelligible fashion and written in standard English?

Reviewer #1: Yes

Reviewer #2: Yes

6. Review Comments to the Author

Reviewer #1: Thank you for addressing my concerns. I have one small comment, I prefer "Future Work" to "Future Works." Using the former makes it seem like there are many more possibilities, while the latter makes it seem like there is a smaller number of things that can be done in the future.

Reviewer #2: The authors have addressed all comments adequately. They have also added more figures and explanations to further clarify some of the raised comments. They have significantly improved the fashion of writing and revised the use of the English language.

7. PLOS authors have the option to publish the peer review history of their article (what does this mean?). If published, this will include your full peer review and any attached files.

Reviewer #1: No

Reviewer #2: No

---

## [Editor Report · Acceptance letter]

12 Nov 2020

PONE-D-20-12174R1 

Automated technique for high-pressure water-based window cleaning and accompanying parametric study 

Dear Dr. Seo:

I'm pleased to inform you that your manuscript has been deemed suitable for publication in PLOS ONE. Congratulations! Your manuscript is now with our production department. 

Kind regards, 

on behalf of

Professor Hongbing Ding 

Academic Editor

PLOS ONE